# Online F-Measure Optimization

**Róbert Busa-Fekete**
Department of Computer Science
University of Paderborn, Germany
busarobi@upb.de

**Balázs Szörényi**
Technion, Haifa, Israel /
MTA-SZTE Research Group on
Artificial Intelligence, Hungary
szorenyibalazs@gmail.com

**Krzysztof Dembczyński**
Institute of Computing Science
Poznań University of Technology, Poland
kdembczynski@cs.put.poznan.pl

**Eyke Hüllermeier**
Department of Computer Science
University of Paderborn, Germany
eyke@upb.de

## Abstract

The F-measure is an important and commonly used performance metric for binary prediction tasks. By combining precision and recall into a single score, it avoids disadvantages of simple metrics like the error rate, especially in cases of imbalanced class distributions. The problem of optimizing the F-measure, that is, of developing learning algorithms that perform optimally in the sense of this measure, has recently tackled by several authors. In this paper, we study the problem of F-measure maximization in the setting of online learning. We propose an efficient online algorithm and provide a formal analysis of its convergence properties. Moreover, first experimental results are presented, showing that our method performs well in practice.

## 1  Introduction

Being rooted in information retrieval [16], the so-called F-measure is nowadays routinely used as a performance metric in various prediction tasks. Given predictions $\widehat{\boldsymbol{y}} = (\widehat{y}_1, \ldots, \widehat{y}_t) \in \{0,1\}^t$ of $t$ binary labels $\boldsymbol{y} = (y_1, \ldots, y_t)$, the F-measure is defined as

$$F(\boldsymbol{y}, \widehat{\boldsymbol{y}}) = \frac{2\sum_{i=1}^{t} y_i \widehat{y}_i}{\sum_{i=1}^{t} y_i + \sum_{i=1}^{t} \widehat{y}_i} = \frac{2 \cdot \text{precision}(\boldsymbol{y}, \widehat{\boldsymbol{y}}) \cdot \text{recall}(\boldsymbol{y}, \widehat{\boldsymbol{y}})}{\text{precision}(\boldsymbol{y}, \widehat{\boldsymbol{y}}) + \text{recall}(\boldsymbol{y}, \widehat{\boldsymbol{y}})} \in [0,1] \ , \qquad (1)$$

where $\text{precision}(\boldsymbol{y}, \widehat{\boldsymbol{y}}) = \sum_{i=1}^{t} y_i \widehat{y}_i / \sum_{i=1}^{t} \widehat{y}_i$, $\text{recall}(\boldsymbol{y}, \widehat{\boldsymbol{y}}) = \sum_{i=1}^{t} y_i \widehat{y}_i / \sum_{i=1}^{t} y_i$, and where $0/0 = 1$ by definition. Compared to measures like the error rate in binary classification, maximizing the F-measure enforces a better balance between performance on the minority and majority class; therefore, it is more suitable in the case of imbalanced data. Optimizing for such an imbalanced measure is very important in many real-world applications where positive labels are significantly less frequent than negative ones. It can also be generalized to a weighted harmonic average of precision and recall. Yet, for the sake of simplicity, we stick to the unweighted mean, which is often referred to as the F1-score or the F1-measure.

Given the importance and usefulness of the F-measure, it is natural to look for learning algorithms that perform optimally in the sense of this measure. However, optimizing the F-measure is a quite challenging problem, especially because the measure is not decomposable over the binary predictions. This problem has received increasing attention in recent years and has been tackled by several authors [19, 20, 18, 10, 11]. However, most of this work has been done in the standard setting of batch learning.

In this paper, we study the problem of F-measure optimization in the setting of online learning [4, 2], which is becoming increasingly popular in machine learning. In fact, there are many applications in which training data is arriving progressively over time, and models need to be updated and maintained incrementally. In our setting, this means that in each round $t$ the learner first outputs a prediction $\widehat{y}_t$ and then observes the true label $y_t$. Formally, the protocol in round $t$ is as follows:

1. first an instance $\boldsymbol{x}_t \in \mathcal{X}$ is observed by the learner,
2. then the predicted label $\widehat{y}_t$ for $\boldsymbol{x}_t$ is computed on the basis of the first $t$ instances $(\boldsymbol{x}_1, \ldots, \boldsymbol{x}_t)$, the $t-1$ labels $(y_1, \ldots, y_{t-1})$ observed so far, and the corresponding predictions $(\widehat{y}_1, \ldots, \widehat{y}_{t-1})$,
3. finally, the label $y_t$ is revealed to the learner.

The goal of the learner is then to maximize

$$F_{(t)} = F((y_1, \ldots, y_t), (\widehat{y}_1, \ldots, \widehat{y}_t)) \tag{2}$$

over time. Optimizing the F-measure in an online fashion is challenging mainly because of the non-decomposability of the measure, and the fact that the $\widehat{y}_t$ cannot be changed after round $t$.

As a potential application of online F-measure optimization consider the recommendation of news from RSS feeds or tweets [1]. Besides, it is worth mentioning that online methods are also relevant in the context of big data and large-scale learning, where the volume of data, despite being finite, prevents from processing each data point more than once [21, 7]. Treating the data as a stream, online algorithms can then be used as single-pass algorithms. Note, however, that single-pass algorithms are evaluated only at the end of the training process, unlike online algorithms that are supposed to learn and predict simultaneously.

We propose an online algorithm for F-measure optimization, which is not only very efficient but also easy to implement. Unlike other methods, our algorithm does not require extra validation data for tuning a threshold (that separates between positive and negative predictions), and therefore allows the entire data to be used for training. We provide a formal analysis of the convergence properties of our algorithm and prove its statistical consistency under different assumptions on the learning process. Moreover, first experimental results are presented, showing that our method performs well in practice.

## 2 Formal Setting

In this paper, we consider a stochastic setting in which $(\boldsymbol{x}_1, y_1), \ldots, (\boldsymbol{x}_t, y_t)$ are assumed to be i.i.d. samples from some unknown distribution $\rho(\cdot)$ on $\mathcal{X} \times \mathcal{Y}$, where $\mathcal{Y} = \{0, 1\}$ is the label space and $\mathcal{X}$ is some instance space. We denote the marginal distribution of the feature vector $X$ by $\mu(\cdot)$.[1] Then, the posterior probability of the positive class, i.e., the conditional probability that $Y = 1$ given $X = \boldsymbol{x}$, is $\eta(\boldsymbol{x}) = \mathbf{P}(Y = 1 \mid X = \boldsymbol{x}) = \frac{\rho(\boldsymbol{x}, 1)}{\rho(\boldsymbol{x}, 0) + \rho(\boldsymbol{x}, 1)}$. The prior distribution of class 1 can be written as $\pi_1 = \mathbf{P}(Y = 1) = \int_{\boldsymbol{x} \in \mathcal{X}} \eta(\boldsymbol{x}) \, \mathrm{d}\mu(\boldsymbol{x})$.

Let $\mathcal{B} = \{f : \mathcal{X} \longrightarrow \{0, 1\}\}$ be the set of all binary classifiers over the set $\mathcal{X}$. The F-measure of a binary classifier $f \in \mathcal{B}$ is calculated as

$$F(f) = \frac{2 \int_{\mathcal{X}} \eta(\boldsymbol{x}) f(\boldsymbol{x}) \, \mathrm{d}\mu(\boldsymbol{x})}{\int_{\mathcal{X}} \eta(\boldsymbol{x}) \, \mathrm{d}\mu(\boldsymbol{x}) + \int_{\mathcal{X}} f(\boldsymbol{x}) \, \mathrm{d}\mu(\boldsymbol{x})} = \frac{2\mathbf{E}\left[\eta(X) f(X)\right]}{\mathbf{E}\left[\eta(X)\right] + \mathbf{E}\left[f(X)\right]}.$$

According to [19], the expected value of (1) converges to $F(f)$ with $t \to \infty$ when $f$ is used to calculate $\widehat{y}$, i.e., $\widehat{y}_t = f(\boldsymbol{x}_t)$. Thus, $\lim_{t \to \infty} \mathbf{E}\left[F((y_1, \ldots, y_t), (f(\boldsymbol{x}_1), \ldots, f(\boldsymbol{x}_t)))\right] = F(f)$.

Now, let $\mathcal{G} = \{g : \mathcal{X} \longrightarrow [0, 1]\}$ denote the set of all *probabilistic* binary classifiers over the set $\mathcal{X}$, and let $\mathcal{T} \subseteq \mathcal{B}$ denote the set of binary classifiers that are obtained by thresholding a classifier $g \in \mathcal{G}$—that is, classifiers of the form

$$g^\tau(\boldsymbol{x}) = [\![g(\boldsymbol{x}) \geq \tau]\!] \tag{3}$$

for some threshold $\tau \in [0, 1]$, where $[\![\cdot]\!]$ is the indicator function that evaluates to 1 if its argument is true and 0 otherwise.

According to [19], the optimal F-score computed as $\max_{f \in \mathcal{B}} F(f)$ can be achieved by a thresholded classifier. More precisely, let us define the *thresholded F-measure* as

$$F(\tau) = F(\eta^\tau) = \frac{2 \int_{\mathcal{X}} \eta(\boldsymbol{x}) [\![\eta(\boldsymbol{x}) \geq \tau]\!] \, \mathrm{d}\mu(\boldsymbol{x})}{\int_{\mathcal{X}} \eta(\boldsymbol{x}) \, \mathrm{d}\mu(\boldsymbol{x}) + \int_{\mathcal{X}} [\![\eta(\boldsymbol{x}) \geq \tau]\!] \, \mathrm{d}\mu(\boldsymbol{x})} = \frac{2\mathbf{E}\left[\eta(X) [\![\eta(X) \geq \tau]\!]\right]}{\mathbf{E}\left[\eta(X)\right] + \mathbf{E}\left[[\![\eta(X) \geq \tau]\!]\right]} \quad (4)$$

Then the *optimal threshold* $\tau^*$ can be obtained as

$$\tau^* = \underset{0 \leq \tau \leq 1}{\operatorname{argmax}} F(\tau) \; . \quad (5)$$

Clearly, for the classifier in the form of (3) with $g(\boldsymbol{x}) = \eta(\boldsymbol{x})$ and $\tau = \tau^*$, we have $F(g^\tau) = F(\tau^*)$. Then, as shown by [19] (see their Theorem 4), the performance of any binary classifier $f \in \mathcal{B}$ cannot exceed $F(\tau^*)$, i.e., $F(f) \leq F(\tau^*)$ for all $f \in \mathcal{B}$. Therefore, estimating posteriors first and adjusting a threshold afterward appears to be a reasonable strategy. In practice, this seems to be the most popular way of maximizing the F-measure in a batch mode; we call it the 2-stage F-measure maximization approach, or 2S for short. More specifically, the 2S approach consists of two steps: first, a classifier is trained for estimating the posteriors, and second, a threshold is tuned on the posterior estimates. For the time being, we are not interested in the training of this classifier but focus on the second step, that is, the labelling of instances via thresholding posterior probabilities. For doing this, suppose a finite set $\mathcal{D}_N = \{(\boldsymbol{x}_i, y_i)\}_{i=1}^N$ of labeled instances are given as training information. Moreover, suppose estimates $\widehat{p}_i = g(\boldsymbol{x}_i)$ of the posterior probabilities $p_i = \eta(\boldsymbol{x}_i)$ are provided by a classifier $g \in \mathcal{G}$. Next, one might define the F-score obtained by applying the threshold classifier $g^\tau$ on the data $\mathcal{D}_N$ as follows:

$$F(\tau; g, \mathcal{D}_N) = \frac{\sum_{i=1}^N y_i [\![\tau \leq g(\boldsymbol{x}_i)]\!]}{\sum_{i=1}^N y_i + \sum_{i=1}^N [\![\tau \leq g(\boldsymbol{x}_i)]\!]} \quad (6)$$

In order to find an optimal threshold $\tau_N \in \operatorname{argmax}_{0 \leq \tau \leq 1} F(\tau; g, \mathcal{D}_N)$, it suffices to search the finite set $\{\widehat{p}_1, \ldots, \widehat{p}_N\}$, which requires time $\mathcal{O}(N \log N)$. In [19], it is shown that $F(\tau; g, \mathcal{D}_N) \xrightarrow{P} F(g^\tau)$ as $N \to \infty$ for any $\tau \in (0, 1)$, and [11] provides an even stronger result: If a classifier $g_{\mathcal{D}_N}$ is induced from $\mathcal{D}_N$ by an $L_1$-consistent learner,[2] and a threshold $\tau_N$ is obtained by maximizing (6) on an independent set $\mathcal{D}'_N$, then $F(g_{\mathcal{D}_N}^{\tau_N}) \xrightarrow{P} F(\tau^*)$ as $N \longrightarrow \infty$ (under mild assumptions on the data distribution).

## 3 Maximizing the F-Measure on a Population Level

In this section we assume that the data distribution is known. According to the analysis in the previous section, optimizing the F-measure boils down to finding the optimal threshold $\tau^*$. At this point, an observation is in order.

**Remark 1.** *In general, the function $F(\tau)$ is neither convex nor concave. For example, when $\mathcal{X}$ is finite, then the denominator and enumerator of (4) are step functions, whence so is $F(\tau)$. Therefore, gradient methods cannot be applied for finding $\tau^*$.*

Nevertheless, $\tau^*$ can be found based on a recent result of [20], who show that finding the root of

$$h(\tau) = \int_{\boldsymbol{x} \in \mathcal{X}} \max\left(0, \eta(\boldsymbol{x}) - \tau\right) \mathrm{d}\mu(\boldsymbol{x}) - \tau \pi_1 \quad (7)$$

is a necessary and sufficient condition for optimality. Note that $h(\tau)$ is continuous and strictly decreasing, with $h(0) = \pi_1$ and $h(1) = -\pi_1$. Therefore, $h(\tau) = 0$ has a unique solution which is $\tau^*$. Moreover, [20] also prove an interesting relationship between the optimal threshold and the F-measure induced by that threshold: $F(\tau^*) = 2\tau^*$.

The marginal distribution of the feature vectors, $\mu(\cdot)$, induces a distribution $\zeta(\cdot)$ on the posteriors: $\zeta(p) = \int_{\boldsymbol{x} \in \mathcal{X}} [\![\eta(\boldsymbol{x}) = p]\!] \, \mathrm{d}\mu(\boldsymbol{x})$ for all $p \in [0, 1]$. By definition, $[\![\eta(\boldsymbol{x}) = p]\!]$ is the Radon-Nikodym derivative of $\frac{\mathrm{d}\mu}{\mathrm{d}\zeta}$, and $\zeta(p)$ the density of observing an instance $\boldsymbol{x}$ for which the probability of the

positive label is $p$. We shall write concisely $\mathrm{d}\nu(p) = \zeta(p)\,\mathrm{d}p$. Since $\nu(\cdot)$ is an *induced probability measure*, the measurable transformation allows us to rewrite the notions introduced above in terms of $\nu(\cdot)$ instead of $\mu(\cdot)$—see, for example, Section 1.4 in [17]. For example, the prior probability $\int_{\mathcal{X}} \eta(\boldsymbol{x})\,\mathrm{d}\mu$ can be written equivalently as $\int_0^1 p\,\mathrm{d}\nu(p)$. Likewise, (7) can be rewritten as follows:

$$h(\tau) = \int_0^1 \max\left(0, p - \tau\right)\mathrm{d}\nu(p) - \tau \int_0^1 p\,\mathrm{d}\nu(p) = \int_\tau^1 p - \tau\,\mathrm{d}\nu(p) - \tau \int_0^1 p\,\mathrm{d}\nu(p)$$

$$= \int_\tau^1 p\,\mathrm{d}\nu(p) - \tau \left[ \int_\tau^1 1\,\mathrm{d}\nu(p) - \int_0^1 p\,\mathrm{d}\nu(p) \right] \tag{8}$$

Equation (8) will play a central role in our analysis. Note that precise knowledge of $\nu(\cdot)$ suffices to find the maxima of $F(\tau)$. This is illustrated by two examples presented in Appendix E, in which we assume specific distributions for $\nu(\cdot)$, namely uniform and Beta distributions.

## 4 Algorithmic Solution

In this section, we provide an algorithmic solution to the online F-measure maximization problem. For this, we shall need in each round $t$ some classifier $g_t \in \mathcal{G}$ that provides us with some estimate $\widehat{p}_t = g_t(\boldsymbol{x}_t)$ of the probability $\eta(\boldsymbol{x}_t)$. We would like to stress again that the focus of our analysis is on optimal thresholding instead of classifier learning. Thus, we assume the sequence of classifiers $g_1, g_2, \ldots$ to be produced by an external online learner, for example, logistic regression trained by stochastic gradient descent.

As an aside, we note that F-measure maximization is not directly comparable with the task that is most often considered and analyzed in online learning, namely regret minimization [4]. This is mainly because the F-measure is a non-decomposable performance metric. In fact, the cumulative regret is a summation of a per-round regret $r_t$, which only depends on the prediction $\widehat{y}_t$ and the true outcome $y_t$ [11]. In the case of the F-measure, the score $F_{(t)}$, and therefore the optimal prediction $\widehat{y}_t$, depends on the entire history, that is, all observations and decisions made by the learner till time $t$. This is discussed in more detail in Section 6.

The most naive way of forecasting labels is to implement online learning as repeated batch learning, that is, to apply a batch learner (such as 2S) to $\mathcal{D}_t = \{(\boldsymbol{x}_i, y_i)\}_{i=1}^t$ in each time step $t$. Obviously, however, this strategy is prohibitively expensive, as it requires storage of all data points seen so far (at least in mini-batches), as well as optimization of the threshold $\tau_t$ and re-computation of the classifier $g_t$ on an ever growing number of examples.

In the following, we propose a more principled technique to maximize the online F-measure. Our approach is based on the observation that $h(\tau^*) = 0$ and $h(\tau)(\tau - \tau^*) < 0$ for any $\tau \in [0,1]$ such that $\tau \neq \tau^*$ [20]. Moreover, it is a monotone decreasing continuous function. Therefore, finding the optimal threshold $\tau^*$ can be viewed as a root finding problem. In practice, however, $h(\tau)$ is not known and can only be estimated. Let us define $h(\tau, y, \widehat{y}) = y\widehat{y} - \tau(y + \widehat{y})$. For now, assume $\eta(\boldsymbol{x})$ to be known and write concisely $\widehat{h}(\tau) = h(\tau, y, [\![\eta(\boldsymbol{x}) \geq \tau]\!])$. We can compute the expectation of $\widehat{h}(\tau)$ with respect to the data distribution for a fixed threshold $\tau$ as follows:

$$\mathbf{E}\left[\widehat{h}(\tau)\right] = \mathbf{E}\left[h(\tau, y, [\![\eta(\boldsymbol{x}) \geq \tau]\!])\right] = \mathbf{E}\left[y\,[\![\eta(\boldsymbol{x}) \geq \tau]\!] - \tau\left(y + [\![\eta(\boldsymbol{x}) \geq \tau]\!]\right)\right]$$

$$= \int_0^1 p\,[\![p \geq \tau]\!]\,\mathrm{d}\nu(p) - \tau\left( \int_0^1 p + [\![p \geq \tau]\!]\,\mathrm{d}\nu(p) \right)$$

$$= \int_\tau^1 p\,\mathrm{d}\nu(p) - \tau\left[ \int_0^1 p\,\mathrm{d}\nu(p) + \int_\tau^1 1\,\mathrm{d}\nu(p) \right] = h(\tau) \tag{9}$$

Thus, an unbiased estimate of $h(\tau)$ can be obtained by evaluating $\widehat{h}(\tau)$ for an instance $\boldsymbol{x}$. This suggests designing a stochastic approximation algorithm that is able to find the root of $h(\cdot)$ similarly to the Robbins-Monro algorithm [12]. Exploiting the relationship between the optimal F-measure and the optimal threshold, $F(\tau^*) = 2\tau^*$, we define the threshold in time step $t$ as

$$\tau_t = \frac{1}{2}F_{(t)} = \frac{a_t}{b_t} \quad \text{where} \quad a_t = \sum_{i=1}^t y_i\widehat{y}_i, \quad b_t = \sum_{i=1}^t y_i + \sum_{i=1}^t \widehat{y}_i \ . \tag{10}$$

With this threshold, the first differences between thresholds, i.e. $\tau_{t+1} - \tau_t$, can be written as follows.

**Proposition 2.** *If thresholds $\tau_t$ are defined according to (10) and $\widehat{y}_{t+1}$ as $[\![\eta(\boldsymbol{x}_{t+1}) > \tau_t]\!]$, then*

$$(\tau_{t+1} - \tau_t)b_{t+1} = h(\tau_t, y_{t+1}, \widehat{y}_{t+1}) \ . \tag{11}$$

The proof of Prop. 2 is deferred to Appendix A. According to (11), the method we obtain "almost" coincides with the update rule of the Robbins-Monro algorithm. There are, however, some notable differences. In particular, the sequence of coefficients, namely the values $1/b_{t+1}$, does not consist of predefined real values converging to zero (as fast as $1/t$). Instead, it consists of random quantities that depend on the history, namely the observed labels $y_1, \ldots, y_t$ and the predicted labels $\widehat{y}_1, \ldots, \widehat{y}_t$. Moreover, these "coefficients" are not independent of $h(\tau_t, y_{t+1}, \widehat{y}_{t+1})$ either. In spite of these additional difficulties, we shall present a convergence analysis of our algorithm in the next section.

The pseudo-code of our online F-measure optimization algorithm, called Online F-measure Optimizer (OFO), is shown in Algorithm 1. The forecast rule can be written in the form of $\widehat{y}_t = [\![p_t \geq \tau_{t-1}]\!]$ for $\boldsymbol{x}_t$ where the threshold is defined in (10) and $p_t = \eta(\boldsymbol{x}_t)$. In practice, we use $\widehat{p}_t = g_{t-1}(\boldsymbol{x}_t)$ as an estimate of the true posterior $p_t$. In line 8 of the code, an online learner $\mathcal{A} : \mathcal{G} \times \mathcal{X} \times \mathcal{Y} \longrightarrow \mathcal{G}$ is assumed, which produces classifiers $g_t$ by incrementally updating the current classifier with the newly observed example, i.e., $g_t = \mathcal{A}(g_{t-1}, \boldsymbol{x}_t, y_t)$.

---
**Algorithm 1** OFO
---
1: Select $g_0$ from $\mathcal{B}$, and set $\tau_0 = 0$
2: **for** $t = 1 \to \infty$ **do**
3:      Observe the instance $\boldsymbol{x}_t$
4:      $\widehat{p}_t \leftarrow g_{t-1}(\boldsymbol{x}_t)$        ▷ estimate posterior
5:      $\widehat{y}_t \leftarrow [\![\widehat{p}_t \geq \tau_{t-1}]\!]$     ▷ current prediction
6:      Observe label $y_t$
7:      Calculate $F_{(t)} = \frac{2a_t}{b_t}$ and $\tau_t = \frac{a_t}{b_t}$
8:      $g_t \leftarrow \mathcal{A}(g_{t-1}, \boldsymbol{x}_t, y_t)$ ▷ update the classifier
9: **return** $\tau_T$
---

In our experimental study, we shall test and compare various state-of-the-art online learners as possible choices for $\mathcal{A}$.

## 5 Consistency

In this section, we provide an analysis of the online F-measure optimizer proposed in the previous section. More specifically, we show the statistical consistency of the OFO algorithm: The sequence of online thresholds and F-scores produced by this algorithm converge, respectively, to the optimal threshold $\tau^*$ and the optimal thresholded F-score $F(\tau^*)$ in probability. As a first step, we prove this result under the assumption of knowledge about the true posterior probabilities; then, in a second step, we consider the case of estimated posteriors.

**Theorem 3.** *Assume the posterior probabilities $p_t = \eta(\boldsymbol{x}_t)$ of the positive class to be known in each step of the online learning process. Then, the sequences of thresholds $\tau_t$ and online F-scores $F_{(t)}$ produced by OFO both converge in probability to their optimal values $\tau^*$ and $F(\tau^*)$, respectively: For any $\epsilon > 0$, we have $\lim_{t\to\infty} \mathbf{P}\big(|\tau_t - \tau^*| > \epsilon\big) = 0$ and $\lim_{t\to\infty} \mathbf{P}\big(|F_{(t)} - F(\tau^*)| > \epsilon\big) = 0$.*

Here is a sketch of the proof of this theorem, the details of which can be found in the supplementary material (Appendix B):

- We focus on $\{\tau_t\}_{t=1}^{\infty}$, which is a stochastic process the filtration of which is defined as $\mathcal{F}_t = \{y_1, \ldots, y_t, \widehat{y}_1, \ldots, \widehat{y}_t\}$. For this filtration, one can show that $\widehat{h}(\tau_t)$ is $\mathcal{F}_t$-measurable and $\mathbf{E}\left[\widehat{h}(\tau_t)|\mathcal{F}_t\right] = h(\tau_t)$ based on (9).

- As a first step, we can decompose the update rule given in (11) as follows: $\mathbf{E}\left[\frac{1}{b_{t+1}}\widehat{h}(\tau_t)\Big|\mathcal{F}_t\right] = \frac{1}{b_t+2}h(\tau_t) + \mathcal{O}\left(\frac{1}{b_t^2}\right)$ conditioned on the filtration $\mathcal{F}_t$ (see Lemma 7).

- Next, we show that the sequence $1/b_t$ behaves similarly to $1/t$, in the sense that $\sum_{t=1}^{\infty} \mathbf{E}\left[1/b_t^2\right] < \infty$ (see Lemma 8). Moreover, one can show that $\sum_{t=1}^{\infty} \mathbf{E}\left[1/b_t\right] \geq \sum_{t=1}^{\infty} \frac{1}{2t} = \infty$.

- Although $h(\tau)$ is not differentiable on $[0, 1]$ in general (it can be piecewise linear, for example), one can show that its finite difference is between $-1 - \pi_1$ and $-\pi_1$ (see Proposition 9 in the appendix). As a consequence of this result, our process defined in (11) does not get stuck even close to $\tau^*$.

- The main part of the proof is devoted to analyzing the properties of the sequence of $\beta_t = \mathbf{E}\left[(\tau_t - \tau^*)^2\right]$ for which we show that $\lim_{t\to\infty} \beta_t = 0$, which is sufficient for the statement

of the theorem. Our proof follows the convergence analysis of [12]. Nevertheless, our analysis essentially differs from theirs, since in our case, the coefficients cannot be chosen freely. Instead, as explained before, they depend on the labels observed and predicted so far. In addition, the noisy estimation of $h(\cdot)$ depends on the labels, too, but the decomposition step allows us to handle this undesired effect.

**Remark 4.** *In principle, the Robbins-Monro algorithm can be applied for finding the root of $h(\cdot)$ as well. This yields an update rule similar to (11), with $1/b_{t+1}$ replaced by $C/t$ for a constant $C > 0$. In this case, however, the convergence of the online F-measure is difficult to analyze (if at all), because the empirical process cannot be written in a nice form. Moreover, as it has been found in the analysis, the coefficient $C$ should be set $\approx 1/\pi_1$ (see Proposition 9 and the choice of $\{k_t\}$ at the end of the proof of Theorem 3). Yet, since $\pi_1$ is not known beforehand, it needs to be estimated from the samples, which implies that the coefficients are not independent of the noisy evaluations of $h(\cdot)$—just like in the case of the* OFO *algorithm. Interestingly,* OFO *seems to properly adjust the values $1/b_{t+1}$ in an adaptive manner ($b_t$ is a sum of two terms, the first of which is $t\pi_1$ in expectation), which is a very nice property of the algorithm. Empirically, based on synthetic data, we found the performance of the original Robbins-Monro algorithm to be on par with OFO.*

As already announced, we are now going to relax the assumption of known posterior probabilities $p_t = \eta(\boldsymbol{x}_t)$. Instead, estimates $\widehat{p}_t = g_t(\boldsymbol{x}_t) \approx p_t$ of these probabilities are obtained by classifiers $g_t$ that are provided by the external online learner in Algorithm 1. More concretely, assume an online learner $\mathcal{A} : \mathcal{G} \times \mathcal{X} \times \mathcal{Y} \longrightarrow \mathcal{G}$, where $\mathcal{G}$ is the set of probabilistic classifiers. Given a current model $g_t$ and a new example $(\boldsymbol{x}_t, y_t)$, this learner produces an updated classifier $g_{t+1} = \mathcal{A}(g_t, \boldsymbol{x}_t, y_t)$. Showing a consistency result for this scenario requires some assumptions on the online learner. With this formal definition of online learner, a statistical consistency result similar to Theorem 3 can be shown. The proof of the following theorem is again deferred to supplementary material (Appendix C).

**Theorem 5.** *Assume that the classifiers $(g_t)_{t=1}^{\infty}$ in the* OFO *framework are provided by an online learner for which the following holds: There is a $\lambda > 0$ such that $\mathbf{E}\left[\int_{\boldsymbol{x} \in \mathcal{X}} |\eta(\boldsymbol{x}) - g_t(\boldsymbol{x})| \, \mathrm{d}\mu(\boldsymbol{x})\right] = \mathcal{O}(t^{-\lambda})$. Then, $F_{(t)} \xrightarrow{P} F(\tau^*)$ and $\tau_t \xrightarrow{P} \tau^*$.*

This theorem's requirement on the online learner is stronger than what is assumed by [11] and recalled in Footnote 2. First, the learner is trained online and not in a batch mode. Second, we also require that the $L_1$ error of the learner goes to 0 with a convergence rate of order $t^{-\lambda}$.

It might be interesting to note that a universal rate of convergence cannot be established without assuming regularity properties of the data distribution, such as smoothness via absolute continuity. Results of that kind are beyond the scope of this study. Instead, we refer the reader to [5, 6] for details on $L_1$ consistency and its connection to the rate of convergence.

## 6   Discussion

*Regret optimization and stochastic approximation:* Stochastic approximation algorithms can be applied for finding the optimum of (4) or, equivalently, to find the unique root of (8) based on noisy evaluations—the latter formulation is better suited for the classic version of the Robbins-Monro root finding algorithm [12]. These algorithms are iterative methods whose analysis focuses on the difference of $F(\tau_t)$ from $F(\tau^*)$, where $\tau_t$ denotes the estimate of $\tau^*$ in iteration $t$, whereas our online setting is concerned with the distance of $F((y_1, \ldots, y_t), (\widehat{y}_1, \ldots, \widehat{y}_t))$ from $F(\tau^*)$, where $\widehat{y}_i$ is the prediction for $y_i$ in round $i$. This difference is crucial because $F(\tau_t)$ only depends on $\tau_t$ and in addition, if $\tau_t$ is close to $\tau^*$ then $F(\tau_t)$ is also close to $F(\tau^*)$ (see [19] for concentration properties), whereas in the online F-measure optimization setup, $F((y_1, \ldots, y_t), (\widehat{y}_1, \ldots, \widehat{y}_t))$ can be very different from $F(\tau^*)$ even if the current estimate $\tau_t$ is close to $\tau^*$ in case the number of previous incorrect predictions is large.

In *online learning* and *online optimization* it is common to work with the notion of (cumulative) regret. In our case, this notion could be interpreted either as $\sum_{i=1}^{t} |F((y_1, \ldots, y_i), (\widehat{y}_1, \ldots, \widehat{y}_i)) - F(\tau^*)|$ or as $\sum_{i=1}^{t} |y_i - \widehat{y}_i|$. After division by $t$, the former becomes the average accuracy of the F-measure over time and the latter the accuracy of our predictions. The former is hard to interpret because $|F((y_1, \ldots, y_i), (\widehat{y}_1, \ldots, \widehat{y}_i)) - F(\tau^*)|$ itself is an aggregate measure of our performance

Table 1: Main statistics of the benchmark datasets and one pass F-scores obtained by OFO and 2S methods on various datasets. The bold numbers indicate when the difference is significant between the performance of OFO and 2S methods. The significance level is set to one sigma that is estimated based on the repetitions.

| | | | | Learner: | LogReg | | Pegasos | | Perceptron | |
|---|---|---|---|---|---|---|---|---|---|---|
| Dataset | #instances | #pos | #neg | #features | OFO | 2S | OFO | 2S | OFO | 2S |
| gisette | 7000 | 3500 | 3500 | 5000 | 0.954 | 0.955 | 0.950 | 0.935 | 0.935 | 0.920 |
| news20.bin | 19996 | 9997 | 9999 | 1355191 | 0.879 | 0.876 | 0.879 | 0.883 | 0.908 | 0.930 |
| Replab | 45671 | 10797 | 34874 | 353754 | 0.924 | 0.923 | 0.926 | 0.928 | 0.914 | 0.914 |
| WebspamUni | 350000 | 212189 | 137811 | 254 | 0.912 | **0.918** | **0.914** | 0.910 | 0.927 | 0.912 |
| epsilon | 500000 | 249778 | 250222 | 2000 | 0.878 | 0.872 | 0.884 | 0.886 | 0.862 | 0.872 |
| covtype | 581012 | 297711 | 283301 | 54 | 0.761 | 0.762 | 0.754 | 0.760 | 0.732 | 0.719 |
| url | 2396130 | 792145 | 1603985 | 3231961 | 0.962 | 0.963 | 0.951 | 0.950 | 0.971 | 0.972 |
| SUSY | 5000000 | 2287827 | 2712173 | 18 | 0.762 | 0.762 | **0.754** | 0.745 | 0.710 | 0.720 |
| kdda | 8918054 | 7614730 | 1303324 | 20216830 | **0.927** | 0.926 | 0.921 | **0.926** | 0.913 | **0.927** |
| kddb | 20012498 | 17244034 | 2768464 | 29890095 | 0.934 | 0.934 | **0.930** | 0.929 | 0.923 | 0.928 |

over the first $t$ rounds, which thus makes no sense to aggregate again. The latter, on the other hand, differs qualitatively from our ultimate goal; in fact, $|F((y_1, \ldots, y_t), (\widehat{y}_1, \ldots, \widehat{y}_t)) - F(\tau^*)|$ is the alternate measure that we are aiming to optimize for instead of the accuracy.

*Online optimization of non-decomposable measures:* Online optimization of the F-measure can be seen as a special case of optimizing non-decomposable loss functions as recently considered by [9]. Their framework essentially differs from ours in several points. First, regarding the data generation process, the adversarial setup with oblivious adversary is assumed, unlike our current study where a stochastic setup is assumed. From this point of view, their assumption is more general since the oblivious adversary captures the stochastic setup. Second, the set of classifiers is restricted to differentiable parametric functions, which may not include the F-measure maximizer. Therefore, their proof of vanishing regret does in general not imply convergence to the optimal F-score. Seen from this point of view, their result is weaker than our proof of consistency (i.e., convergence to the optimal F-measure in probability if the posterior estimates originate from a consistent learner). There are some other non-decomposable performance measures which are intensively used in many practical applications. Their optimization had already been investigated in the online or one-pass setup. The most notable such measure might be the area under the ROC curve (AUC) which had been investigated in an online learning framework by [21, 7].

## 7   Experiments

In this section, the performance of the OFO algorithm is evaluated in a one-pass learning scenario on benchmark datasets, and compared with the performance of the 2-stage F-measure maximization approach (2S) described in Section 2. We also assess the rate of convergence of the OFO algorithm in a pure online learning setup.[3]

The online learner $\mathcal{A}$ in OFO was implemented in different ways, using Logistic Regression (LOGREG), the classical Perceptron algorithm (PERCEPTRON) [13] and an online linear SVM called PEGASOS [14]. In the case of LOGREG, we applied the algorithm introduced in [15] which handles L1 and L2 regularization. The hyperparameters of the methods and the validation procedures are described below and in more detail in Appendix D. If necessary, the raw outputs of the learners were turned into probability estimates, i.e., they were rescaled to $[0, 1]$ using logistic transform.

We used in the experiments nine datasets taken from the LibSVM repository of binary classification tasks.[4] Many of these datasets are commonly used as benchmarks in information retrieval where the F-score is routinely applied for model selection. In addition, we also used the textual data released in the Replab challenge of identifying relevant tweets [1]. We generated the features used by the winner team [8]. The main statistics of the datasets are summarized in Table 1.

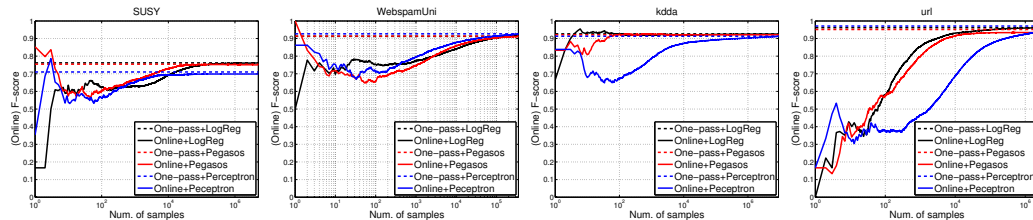

Figure 1: Online F-scores obtained by OFO algorithm on various dataset. The dashed lines represent the one-pass performance of the OFO algorithm from Table 1 which we considered as baseline.

**One-pass learning.** In one-pass learning, the learner is allowed to read the training data only once, whence online learners are commonly used in this setting. We run OFO along with the three classifiers trained on $80\%$ of the data. The learner obtained by OFO is of the form $g_t^{\tau_t}$, where $t$ is the number of training samples. The rest $20\%$ of the data was used to evaluate $g_t^{\tau_t}$ in terms of the F-measure. We run every method on 10 randomly shuffled versions of the data and averaged results. The means of the F-scores computed on the test data are shown in Table 1. As a baseline, we applied the 2S approach. More concretely, we trained the same set of learners on $60\%$ of the data and validated the threshold on $20\%$ by optimizing (6). Since both approaches are consistent, the performance of OFO should be on par with the performance of 2S. This is confirmed by the results, in which significant differences are observed in only 7 of 30 cases. These differences in performance might be explained by the finiteness of the data. The advantage of our approach over 2S is that there is no need of validation and the data needs to be read only once, therefore it can be applied in a pure one-pass learning scenario. The hyperparameters of the learning methods are chosen based on the performance of 2S. We tuned the hyperparameters in a wide range of values which we report in Appendix D.

**Online learning.** The OFO algorithm has also been evaluated in the online learning scenario in terms of the online F-measure (2). The goal of this experiment is to assess the convergence rate of OFO. Since the optimal F-measure is not known for the datasets, we considered the test F-scores reported in Table 1. The results are plotted in Figure 1 for four benchmark datasets (the plots for the remaining datasets can be found in Appendix G). As can be seen, the online F-score converges to the test F-score obtained in one-pass evalaution in almost every case. There are some exceptions in the case of PEGASOS and PERCEPTRON. This might be explained by the fact that SVM-based methods as well as the PERCEPTRON tend to produce poor probability estimates in general (which is a main motivation for calibration methods turning output scores into valid probabilities [3]).

## 8 Conclusion and Future Work

This paper studied the problem of online F-measure optimization. Compared to many conventional online learning tasks, this is a specifically challenging problem, mainly because of the non-decomposable nature of the F-measure. We presented a simple algorithm that converges to the optimal F-score when the posterior estimates are provided by a sequence of classifiers whose $L_1$ error converges to zero as fast as $t^{-\lambda}$ for some $\lambda > 0$. As a key feature of our algorithm, we note that it is a purely online approach; moreover, unlike approaches such as 2S, there is no need for a hold-out validation set in batch mode. Our promising results from extensive experiments validate the empirical efficacy of our algorithm.

For future work, we plan to extend our online optimization algorithm to a broader family of complex performance measures which can be expressed as ratios of linear combinations of true positive, false positive, false negative and true negative rates [10]; the F-measure also belongs to this family. Moreover, going beyond consistency, we plan to analyze the rate of convergence of our OFO algorithm. This might be doable thanks to several nice properties of the function $h(\tau)$. Finally, an intriguing question is what can be said about the case when some bias is introduced because the classifier $g_t$ does not converge to $\eta$.

**Acknowledgments.** Krzysztof Dembczyński is supported by the Polish National Science Centre under grant no. 2013/09/D/ST6/03917.

## Footnotes

[1]$\mathcal{X}$ is assumed to exhibit the required measurability properties.

[2]A learning algorithm, viewed as a map from samples $\mathcal{D}_N$ to classifiers $g_{\mathcal{D}_N}$, is called $L_1$-consistent w.r.t. the data distribution $\rho$ if $\lim_{N \to \infty} \mathbf{P}_{\mathcal{D}_N \sim \rho} \left( \int_{\boldsymbol{x} \in \mathcal{X}} |g_{\mathcal{D}_N}(\boldsymbol{x}) - \eta(\boldsymbol{x})| \, \mathrm{d}\mu(\boldsymbol{x}) > \epsilon \right) = 0$ for all $\epsilon > 0$.

[3]Additional results of experiments conducted on synthetic data are presented in Appendix F.

[4]`http://www.csie.ntu.edu.tw/~cjlin/libsvmtools/datasets/binary.html`

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
