[Supplementary Material · nips_418_fm_full.pdf]

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

## Footnotes

[1] $\mathcal{X}$ is assumed to exhibit the required measurability properties.

[2]A learning algorithm, viewed as a map from samples $\mathcal{D}_N$ to classifiers $g_{\mathcal{D}_N}$, is called $L_1$-consistent w.r.t. the data distribution $\rho$ if $\lim_{N \to \infty} \mathbf{P}_{\mathcal{D}_N \sim \rho} \left( \int_{\boldsymbol{x} \in \mathcal{X}} |g_{\mathcal{D}_N}(\boldsymbol{x}) - \eta(\boldsymbol{x})| \, \mathrm{d}\mu(\boldsymbol{x}) > \epsilon \right) = 0$ for all $\epsilon > 0$.

[3]Additional results of experiments conducted on synthetic data are presented in Appendix F.

[4]`http://www.csie.ntu.edu.tw/~cjlin/libsvmtools/datasets/binary.html`

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

# Supplementary material for "Online F-Measure Optimization"

## A Proposition 2

For reading convenience we restate Proposition 2.

**Proposition 6.** *If thresholds $\tau_t$ are defined according to (10) and $\widehat{y}_{t+1}$ as $[\![\eta(\boldsymbol{x}_{t+1}) > \tau_t]\!]$, then*

$$(\tau_{t+1} - \tau_t)b_{t+1} = h(\tau_t, y_{t+1}, \widehat{y}_{t+1}) \ .$$

*Proof.* The proof is a simple analysis of the three cases $y_{t+1} = \widehat{y}_{t+1} = 1$, $y_{t+1} \neq \widehat{y}_{t+1}$ and $y_{t+1} = \widehat{y}_{t+1} = 0$:

- If $y_{t+1} = \widehat{y}_{t+1} = 1$, then we have $a_{t+1} = a_t + 1$ and $b_{t+1} = b_t + 2$, and so

$$\tau_{t+1} - \tau_t = \frac{a_t + 1}{b_t + 2} - \frac{a_t}{b_t} = \frac{b_t - 2a_t}{(b_t + 2)b_t} = \frac{b_t - 2a_t}{b_{t+1}b_t}$$

  and

$$h(\tau_t, y_{t+1}, \widehat{y}_{t+1}) = y_t\widehat{y}_t - \tau_t(y_t + \widehat{y}_t) = 1 - 2\tau_t = \frac{b_t - 2a_t}{b_t} = (\tau_{t+1} - \tau_t)b_{t+1}.$$

- If $y_{t+1} \neq \widehat{y}_{t+1} = 1$, then we have $a_{t+1} = a_t$ and $b_{t+1} = b_t + 1$, and so

$$\tau_{t+1} - \tau_t = \frac{a_t}{b_t + 1} - \frac{a_t}{b_t} = -\frac{a_t}{(b_t + 1)b_t} = -\frac{a_t}{b_{t+1}b_t}$$

  and

$$h(\tau_t, y_{t+1}, \widehat{y}_{t+1}) = -\tau_t = -\frac{a_t}{b_t} = (\tau_{t+1} - \tau_t)b_{t+1}.$$

- If $y_{t+1} = \widehat{y}_{t+1} = 0$, then we have $a_{t+1} = a_t$ and $b_{t+1} = b_t$, and so

$$\tau_{t+1} - \tau_t = \frac{a_t}{b_t} - \frac{a_t}{b_t} = 0,$$

  and

$$h(\tau_t, y_{t+1}, \widehat{y}_{t+1}) = 0 = (\tau_{t+1} - \tau_t)b_{t+1},$$

$\square$

## B Consistency with knowledge about the true posterior probabilities

For the proof of Theorem 3 we will need the following two lemmas. As a first step, one might decompose the update rule of the threshold $\tau_t$ given in (11). Based on Lemma 7, the difference sequence of the thresholds conditioned on the filtration can be rewritten such that the terms after the decomposition only depend on the data distribution, but not on the filtration.

**Lemma 7.**

$$\mathbf{E}\left[\frac{1}{b_{t+1}}\widehat{h}(\tau_t)\Big|\mathcal{F}_t\right] = \frac{1}{b_t + 2}h(\tau_t) + \mathcal{O}\left(\frac{1}{b_t^2}\right)$$

*where the $\mathcal{O}(.)$ notation hides only universal constants.*

*Proof.* Simple calculation yields that

$$\mathbf{E}\left[\frac{1}{b_{t+1}}\widehat{h}(\tau_t)|\mathcal{F}_t\right] = \mathbf{E}\left[\frac{1}{b_t + 2}\widehat{h}(\tau_t)\Big|\mathcal{F}_t\right] - \mathbf{E}\left[\left(\frac{1}{b_t + 2} - \frac{1}{b_{t+1}}\right)\widehat{h}(\tau_t)\Big|\mathcal{F}_t\right]$$

$$= \frac{1}{b_t + 2}h(\tau_t) - \mathbf{E}\left[\left(\frac{1}{b_t + 2} - \frac{1}{b_{t+1}}\right)\widehat{h}(\tau_t)\Big|\mathcal{F}_t\right]$$

$$= \frac{1}{b_t + 2}h(\tau_t) + \mathcal{O}\left(\frac{1}{b_t^2}\right) \tag{12}$$

where (12) follows from the fact that

$$\frac{1}{b_t + 2} - \frac{1}{b_{t+1}} = \begin{cases} -\frac{2}{b_t^2 + 2b_t}, & \text{if } y_{t+1} = \widehat{y}_{t+1} = 0 \\ 0, & \text{if } y_{t+1} = \widehat{y}_{t+1} = 1 \\ -\frac{1}{b_t^2 + b_t - 2}, & \text{if } y_{t+1} \neq \widehat{y}_{t+1} \end{cases}$$

and $\widehat{h}(\tau) \geq -1$. This completes the proof. $\square$

In the update rule of the thresholds given in (11), the $1/b_{t+1}$ values controls the step size, and in fact, these values play a similar role like the coefficient in the stochastic approximation algorithm of [12] (the coefficients are denoted by $a_t$ in the original paper). Therefore, the sequence of $1/b_{t+1}$ should behave similar to $1/t$, that is $\sum_{t=1}^{\infty} 1/b_t = \infty$ and $\sum_{t=1}^{\infty} 1/b_t^2 < \infty$. Since $b_t \leq 2t$, we have that $\sum_{t=1}^{\infty} 1/b_t \geq \sum_{t=1}^{\infty} 1/2t = \infty$. The infinite sum of the square of $1/b_{t+1}$ is analysed in the next lemma.

**Lemma 8.**

$$\mathbf{E}\left[\frac{1}{b_t^2}\right] = \mathcal{O}\left(\frac{1}{t^2}\right)$$

*where the $\mathcal{O}(.)$ notation hides only universal constants. Consequently, it holds that*

$$\sum_{t=1}^{\infty} \mathbf{E}\left[\frac{1}{b_t^2}\right] < \infty$$

*Proof.* By applying the Chernoff bound for $c_t = \sum_{t'=1}^{t} y_{t'}$ with $t\pi_1/3$ as error term, we have

$$\mathbf{P}\left(c_t \leq \frac{2}{3}t\pi_1\right) \leq \underbrace{\exp\left(\frac{-2t\pi_1^2}{9}\right)}_{\gamma_t} . \tag{13}$$

Therefore, based on (13), we can upper bound $\mathbf{E}[1/(c_t^2 + 1)]$ as

$$\mathbf{E}\left[\frac{1}{(c_t^2 + 1)}\right] \leq \gamma_t + (1 - \gamma_t)\frac{9}{4t^2\pi_1^2} \leq \gamma_t + \frac{9}{4t^2\pi_1^2} \tag{14}$$

Next, note that $1 \leq 1/b_t^2$ since $\tau_0 = 0$ which results in that $\widehat{y}_1 = 1$. Moreover, $1/b_t^2 \leq 1/(c_t^2 + 1)$ since $b_t \geq (c_t + 1)$ which proofs the first claim.

In order to proof the second claim, we might rewrite

$$\sum_{t=1}^{\infty} \mathbf{E}\left[\frac{1}{b_t^2}\right] \leq \sum_{t=1}^{\infty} \mathbf{E}\left[\frac{1}{c_t^2 + 1}\right]$$

$$\leq \sum_{t=1}^{\infty} \gamma_t + \frac{9}{\pi_1^2 4}\sum_{t=1}^{\infty} \frac{1}{t^2} < \infty \tag{15}$$

where we obtained (15) by applying (14). Based on the definition $\gamma_t$, it readily follows that (15) is finite which completes the proof. $\square$

Before we turn to the proof of the Theorem 3, we show that function $h(\cdot)$ is not "too flat" anywhere. In general, function $h(\cdot)$ might be not differentiable everywhere (for example, it can be piecewise linear, or it might be discontinuous for some $\tau \in [0, 1]$). Nevertheless, its finite difference is strictly negative based on the next proposition.

**Proposition 9.** *For any $\tau, \tau' \in [0, 1]$ such that $\tau \neq \tau'$, it holds*

$$-1 - \pi_1 \leq \frac{h(\tau) - h(\tau')}{\tau - \tau'} \leq -\pi_1 .$$

*Proof.* Based on (8), we can write $h(\tau)$ in the form of

$$h(\tau) = \int_0^1 \max(0, p - \tau)\, d\nu(p) - \tau\pi_1 \ .$$

Next, define $g(\tau, \tau')$ as

$$g(\tau, \tau') = \frac{1}{\tau - \tau'} \int_0^1 \max(0, p - \tau) - \max(0, p - \tau')\, d\nu(p)$$

$$= \int_0^1 \frac{\max(0, p - \tau) - \max(0, p - \tau')}{\tau - \tau'}\, d\nu(p) \ .$$

Simple calculation yields that

$$\frac{h(\tau) - h(\tau')}{\tau - \tau'} = g(\tau, \tau') - \pi_1$$

Now there are two cases:

1. When $\tau < \tau'$, we have

$$\max(0, p - \tau) - \max(0, p - \tau') = \begin{cases} \tau' - \tau & \text{if } p > \tau' > \tau \\ p - \tau & \text{if } \tau' > p > \tau \\ 0 & \text{otherwise} \end{cases}$$

thus

$$\frac{\max(0, p - \tau) - \max(0, p - \tau')}{\tau - \tau'} = \begin{cases} \frac{\tau' - \tau}{\tau - \tau'} = -1 & \text{if } p > \tau' > \tau \\ -1 \leq \frac{p - \tau}{\tau - \tau'} \leq 0 & \text{if } \tau' > p > \tau \\ 0 & \text{otherwise} \end{cases}$$

And so $-1 \leq g(\tau, \tau') \leq 0$, which yields that $-1 - \pi_1 \leq \frac{h(\tau) - h(\tau')}{\tau - \tau'} \leq -\pi_1$ in this case.

2. When $\tau > \tau'$, we have

$$\max(0, p - \tau) - \max(0, p - \tau') = \begin{cases} \tau' - \tau & \text{if } p > \tau > \tau' \\ \tau' - p & \text{if } \tau > p > \tau' \\ 0 & \text{otherwise} \end{cases}$$

thus

$$\frac{\max(0, p - \tau) - \max(0, p - \tau')}{\tau - \tau'} = \begin{cases} \frac{\tau' - \tau}{\tau - \tau'} = -1 & \text{if } p > \tau > \tau' \\ -1 \leq \frac{\tau' - p}{\tau - \tau'} \leq 0 & \text{if } \tau > p > \tau' \\ 0 & \text{otherwise} \end{cases}$$

And similarly to the previous case, we have $-1 \leq g(\tau, \tau') \leq 0$, which yields that $-1 - \pi_1 \leq \frac{h(\tau) - h(\tau')}{\tau - \tau'} \leq -\pi_1$ in this case as well.

This completes the proof. $\qquad\square$

The proof of Theorem 3 follows the proof of [12] given for the stochastic approximation method for root finding. However, our proof differs essentially from theirs, because of the lack of independence between $1/b_{t+1}$ and $\widehat{h}(\tau_t)$.

*Proof. of Theorem 3*
Our goal is to show that $\lim_{t \to \infty} \beta_t = 0$ where

$$\beta_t = \mathbf{E}\left[(\tau_t - \tau^*)^2\right]$$

which yields the convergence of the thresholds in probability, formally, $\tau_t \xrightarrow{P} \tau^*$, and consequently $F_{(t)} \xrightarrow{P} F(\tau^*)$, since $F(\tau^*) = 2\tau^*$ and, in addition, $F_{(t)} = 2\tau_t$ based on the definition of $F_{(t)}$ and $\tau_t$ given in (2) and (10), respectively.

We can decompose $\beta_t$ as follows:

$$
\begin{aligned}
\beta_{t+1} &= \mathbf{E}\left[(\tau_{t+1} - \tau^*)^2\right] \\
&= \mathbf{E}\left[\mathbf{E}\left[(\tau_{t+1} - \tau^*)^2 \middle| \mathcal{F}_t\right]\right] \\
&= \mathbf{E}\left[\mathbf{E}\left[(\tau_t - \tau^* + \frac{1}{b_{t+1}}\widehat{h}(\tau_t))^2 \middle| \mathcal{F}_t\right]\right] \\[4pt]
&= \mathbf{E}\left[\mathbf{E}\left[(\tau_t - \tau^*)^2 \middle| \mathcal{F}_t\right]\right] + 2\mathbf{E}\left[\mathbf{E}\left[(\tau_t - \tau^*)\frac{1}{b_{t+1}}\widehat{h}(\tau_t) \middle| \mathcal{F}_t\right]\right] + \mathbf{E}\left[\mathbf{E}\left[\left(\frac{\widehat{h}(\tau_t)}{b_{t+1}}\right)^2 \middle| \mathcal{F}_t\right]\right] \\[4pt]
&= \beta_t + 2\mathbf{E}\left[\frac{\tau_t - \tau^*}{b_t + 2}h(\tau_t) + \mathcal{O}\left(\frac{1}{b_t^2}\right)\right] + \mathbf{E}\left[\mathbf{E}\left[\left(\frac{\widehat{h}(\tau_t)}{b_{t+1}}\right)^2 \middle| \mathcal{F}_t\right]\right] \\[4pt]
&= \beta_t - 2\mathbf{E}\left[\frac{\tau^* - \tau_t}{b_t + 2}h(\tau_t)\right] + \mathcal{O}\left(\mathbf{E}\left[\frac{1}{b_t^2}\right]\right) + \mathbf{E}\left[\mathbf{E}\left[\left(\frac{\widehat{h}(\tau_t)}{b_{t+1}}\right)^2 \middle| \mathcal{F}_t\right]\right] \\[4pt]
&= \beta_t - 2\mathbf{E}\left[\frac{\tau^* - \tau_t}{b_t + 2}h(\tau_t)\right] + \mathcal{O}\left(\mathbf{E}\left[\frac{1}{b_t^2}\right]\right)
\end{aligned}
$$

$$\text{(16)} \qquad \text{(17)} \qquad \text{(18)}$$

where (16) follows from the update rule defined in (11), and (17) is obtained by applying Lemma 7. Finally, (18) follows from the fact that $\widehat{h}(\tau_t)$ is bounded and $1/(b_t + 2) \le 1/b_{t+1} \le 1/b_t$. Note that $\mathcal{O}$ hides only universal constants, therefore $\mathbf{E}(\mathcal{O}(.)) = \mathcal{O}(\mathbf{E}(.))$. So the error term $\beta_{t+1}$ can be written as

$$\beta_{t+1} = \beta_t - 2\delta_t + \mathcal{O}\left(\mathbf{E}\left[\frac{1}{b_t^2}\right]\right) \tag{19}$$

where

$$\delta_t = \mathbf{E}\left[\frac{\tau^* - \tau_t}{b_t + 2}h(\tau_t)\right] \quad .$$

Summing up (19), we have

$$\beta_{t+1} = \beta_1 - 2\sum_{t'=1}^{t}\delta_{t'} + \sum_{t'=1}^{t}\mathcal{O}\left(\mathbf{E}\left[\frac{1}{b_{t'}^2}\right]\right)$$

Hence $h(\tau)(\tau^* - \tau) > 0$, it holds that $\delta_t > 0$, thus we have

$$
\begin{aligned}
\sum_{t'=1}^{t}\delta_{t'} &\le \frac{1}{2}\left[\beta_1 + \sum_{t'=1}^{t}\mathcal{O}\left(\mathbf{E}\left[\frac{1}{b_{t'}^2}\right]\right)\right] \\
&\le \frac{1}{2}\left[\beta_1 + \sum_{t'=1}^{\infty}\mathcal{O}\left(\mathbf{E}\left[\frac{1}{b_{t'}^2}\right]\right)\right]
\end{aligned}
\tag{20}
$$

Based on Lemma 8, $\sum_{t=1}^{\infty}\mathcal{O}\left(\mathbf{E}\left[\frac{1}{b_t^2}\right]\right)$ is finite. Therefore, $\sum_{t'=1}^{\infty}\delta_{t'}$ is also finite, and so there exists a $\beta \ge 0$, such that

$$\lim_{t\to\infty}\beta_t = \beta_1 - 2\sum_{t'=1}^{\infty}\delta_{t'} + \sum_{t'=1}^{\infty}\mathcal{O}\left(\mathbf{E}\left[\frac{1}{b_{t'}^2}\right]\right) = \beta \quad .$$

since all sums are finite above, and $b \ge 0$ based on (20).

As a next step, we show that $\beta = 0$. For doing this, first note that $\frac{1}{2(t+1)} \le \frac{1}{b_t+2}$ for any $t > 0$, thus it holds

$$\sum_{t=1}^{\infty}\frac{1}{2(t+1)}\mathbf{E}\left[(\tau^* - \tau_t)h(\tau_t)\right] \le \sum_{t=1}^{\infty}\delta_t < \infty$$

Now suppose that there exists a sequence $\{k_t\}$ of non-negative real numbers such that

$$k_t \beta_t \leq \mathbf{E}\left[(\tau^* - \tau_t)h(\tau_t)\right] \tag{21}$$

and

$$\sum_{t=1}^{\infty} \frac{k_t}{2(t+1)} = \infty \ . \tag{22}$$

From (21), we have

$$\sum_{t=1}^{\infty} \frac{k_t}{2(t+1)} \beta_t \leq \sum_{t=1}^{\infty} \frac{1}{2(t+1)} \mathbf{E}\left[(\tau^* - \tau_t)h(\tau_t)\right]$$

$$\leq \sum_{t=1}^{\infty} \delta_t < \infty.$$

From (22), it follows that for any $\epsilon > 0$ there must exist infinitely many values such that $\beta_t < \epsilon$. Since we know that $\lim_{t\to\infty} \beta_t = b$ exists, thus $\beta = 0$ necessarily (if $\{k_n\}$ does exist).

The only thing remained to be shown is the existence of $\{k_n\}$ which satisfies (21) and (22). Based on Proposition 9, we have

$$-\pi_1 \leq \sup_{\tau \neq \tau^*} \left[ \frac{h(\tau) - h(\tau^*)}{\tau - \tau^*} \right] = \sup_{\tau \neq \tau^*} \left[ \frac{h(\tau)}{\tau - \tau^*} \right] \ .$$

So we can lower bound $\mathbf{E}\left[(\tau^* - \tau_t)h(\tau_t)\right]$ as

$$\mathbf{E}\left[(\tau^* - \tau_t)h(\tau_t)\right] = \mathbf{E}\left[(\tau^* - \tau_t)^2 \frac{h(\tau_t)}{\tau^* - \tau_t}\right]$$

$$\geq \pi_1 \mathbf{E}\left[(\tau^* - \tau_t)^2\right]$$

$$\geq \pi_1 \beta_t$$

Therefore the constant sequence $k_t = \pi_1$ satisfies (21) and (22). This completes the proof. $\qquad\square$

## C   Consistency with estimated posterior probabilities

Assuming that the posterior estimate is provided as $p_{t+1} = g_t(\boldsymbol{x}_{t+1})$ in time step $t$, the update rule of the thresholds can be written in the form of

$$\tau_{t+1} - \tau_t = \frac{1}{b_{t+1}} h(\tau_t, y_{t+1}, [\![g_t(\boldsymbol{x}_{t+1}) \geq \tau_t]\!]) \ . \tag{23}$$

Let us write concisely $\tilde{h}(\tau, g) = h(\tau, y, [\![g(\boldsymbol{x}) \geq \tau]\!])$. As a first step, we can upper bound the difference between $\tilde{h}(\tau, g)$ and $\widehat{h}(\tau_t)$ in terms of the $L_1$ error of $g$ with respect to the data distribution as follows.

**Lemma 10.**

$$\left| \mathbf{E}\left[\tilde{h}(\tau_t, g_t) - \widehat{h}(\tau_t) | \mathcal{F}_t\right] \right| \leq \int_{\boldsymbol{x} \in \mathcal{X}} |\eta(\boldsymbol{x}) - g_t(\boldsymbol{x})| \, \mathrm{d}\mu(\boldsymbol{x}) = \|\eta - g_t\|_1$$

*where* $\tilde{h}(\tau, g) = h(\tau, y, [\![g(\boldsymbol{x}) \geq \tau]\!]) = h(\tau, y, g^\tau(\boldsymbol{x}))$.

*Proof.* First, compute

$$\tilde{h}(\tau, g) - \widehat{h}(\tau) = y [\![g(\boldsymbol{x}) \geq \tau]\!] - \tau (y + [\![g(\boldsymbol{x}) \geq \tau]\!]) - y [\![\eta(\boldsymbol{x}) \geq \tau]\!] + \tau (y + [\![\eta(\boldsymbol{x}) \geq \tau]\!])$$

$$= y [\![g(\boldsymbol{x}) \geq \tau]\!] - \tau [\![g(\boldsymbol{x}) \geq \tau]\!] - y [\![\eta(\boldsymbol{x}) \geq \tau]\!] + \tau [\![\eta(\boldsymbol{x}) \geq \tau]\!]$$

$$= y ([\![g(\boldsymbol{x}) \geq \tau]\!] - [\![\eta(\boldsymbol{x}) \geq \tau]\!]) - \tau ([\![g(\boldsymbol{x}) \geq \tau]\!] - [\![\eta(\boldsymbol{x}) \geq \tau]\!])$$

$$= (y - \tau) ([\![g(\boldsymbol{x}) \geq \tau]\!] - [\![\eta(\boldsymbol{x}) \geq \tau]\!])$$

Now, note that

$$\llbracket g(\boldsymbol{x}) \geq \tau \rrbracket - \llbracket \eta(\boldsymbol{x}) \geq \tau \rrbracket = \begin{cases} 1 & \text{if } g(\boldsymbol{x}) > \tau > \eta(\boldsymbol{x}) \\ -1 & \text{if } \eta(\boldsymbol{x}) > \tau > g(\boldsymbol{x}) \end{cases}$$

therefore, we have

$$\left| \tilde{h}(\tau, g) - \widehat{h}(\tau) \right| = |(y - \tau)(\llbracket g(\boldsymbol{x}) \geq \tau \rrbracket - \llbracket \eta(\boldsymbol{x}) \geq \tau \rrbracket)|$$
$$\leq |\eta(\boldsymbol{x}) - g(\boldsymbol{x})| \quad .$$

Consequently, we can upper bound the absolute value of

$$\mathbf{E}\left[ \tilde{h}(\tau_t, g_t) - \widehat{h}(\tau_t)|\mathcal{F}_t, \boldsymbol{x}_{t+1} \right] = (\eta(\boldsymbol{x}_{t+1}) - \tau_t)(\llbracket g(\boldsymbol{x}) \geq \tau \rrbracket - \llbracket \eta(\boldsymbol{x}) \geq \tau \rrbracket)$$

as follows:

$$\left| \mathbf{E}\left[ \tilde{h}(\tau_t, g_t) - \widehat{h}(\tau_t)|\mathcal{F}_t, \boldsymbol{x}_{t+1} \right] \right| \leq |\eta(\boldsymbol{x}_{t+1}) - g_t(\boldsymbol{x}_{t+1})| \quad .$$

Therefore we can upper bound the following $\mathbf{E}\left[ \tilde{h}(\tau_t, g_t) - \widehat{h}(\tau_t)|\mathcal{F}_t \right]$ as

$$\left| \mathbf{E}\left[ \tilde{h}(\tau_t, g_t) - \widehat{h}(\tau_t) \middle| \mathcal{F}_t \right] \right| = \mathbf{E}\left[ \left| \mathbf{E}\left[ \tilde{h}(\tau_t, g_t) - \widehat{h}(\tau_t) \middle| \mathcal{F}_t, \boldsymbol{x}_{t+1} \right] \right| \middle| \mathcal{F}_t \right]$$
$$\leq \mathbf{E}\left[ |\eta(\boldsymbol{x}) - g_t(\boldsymbol{x})| |\mathcal{F}_t \right]$$
$$= \int_{\boldsymbol{x} \in \mathcal{X}} |\eta(\boldsymbol{x}) - g_t(\boldsymbol{x})| \, \mathrm{d}\mu(\boldsymbol{x})$$
$$= \|\eta - g_t\|_1$$

$\square$

We need to following results for decomposing the update rule given 10 similarly to Lemma 7.

**Lemma 11.**
$$\mathbf{E}\left[ \frac{1}{b_{t+1}}(\tilde{h}(\tau_t, g_t) - \widehat{h}(\tau_t)) \middle| \mathcal{F}_t \right] = \frac{1}{b_t + 2}\|\eta - g_t\|_1 + \mathcal{O}\left(\frac{1}{b_t^2}\right)$$

*where the $\mathcal{O}(.)$ notation hides only universal constants.*

*Proof.* Simple calculation yields that

$$\mathbf{E}\left[ \frac{1}{b_{t+1}}(\tilde{h}(\tau_t, g_t) - \widehat{h}(\tau_t))|\mathcal{F}_t \right] = \mathbf{E}\left[ \frac{1}{b_t + 2}(\tilde{h}(\tau_t, g_t) - \widehat{h}(\tau_t)) \middle| \mathcal{F}_t \right]$$
$$- \mathbf{E}\left[ \left(\frac{1}{b_t + 2} - \frac{1}{b_{t+1}}\right)(\tilde{h}(\tau_t, g_t) - \widehat{h}(\tau_t)) \middle| \mathcal{F}_t \right]$$
$$\leq \frac{1}{b_t + 2}\|\eta - g_t\|_1$$
$$- \mathbf{E}\left[ \left(\frac{1}{b_t + 2} - \frac{1}{b_{t+1}}\right)(\tilde{h}(\tau_t, g_t) - \widehat{h}(\tau_t)) \middle| \mathcal{F}_t \right] \quad (24)$$
$$= \frac{1}{b_t + 2}\|\eta - g_t\|_1 + \mathcal{O}\left(\frac{1}{b_t^2}\right) \quad (25)$$

where (24) follows from Lemma 10 and (25) can be computed as (12) in the proof of Lemma 7. This completes the proof. $\square$

The proof of Theorem 5 goes in a similar way like the proof of Theorem 3. The basic difference is that there is a new term in the decomposition of $\beta_{t+1}$ which stems from the error of the classifiers which provides the posterior estimates. But this term can be upper bounded based on Lemma 10, and based on the fact that the $L_1$ error of the learner is vanishing as fast as $t^{-\lambda}$ where $1 > \lambda > 0$.

*Proof.* of Theorem 5

Similarly to the proof of Theorem 3, let us decompose $\beta_{t+1}$ which now depend on $\tilde{h}(\cdot)$. We have

$$
\begin{aligned}
\beta_{t+1} &= \mathbf{E}\left[(\tau_{t+1} - \tau^*)^2\right] \\
&= \mathbf{E}\left[\mathbf{E}\left[(\tau_{t+1} - \tau^*)^2 \middle| \mathcal{F}_t\right]\right] \\
&= \mathbf{E}\left[\mathbf{E}\left[(\tau_t - \tau^* + \frac{1}{b_{t+1}}\tilde{h}(\tau_t))^2 \middle| \mathcal{F}_t\right]\right] \qquad (26) \\
&= \mathbf{E}\left[\mathbf{E}\left[(\tau_t - \tau^*)^2 \middle| \mathcal{F}_t\right]\right] \\
&\qquad + 2\mathbf{E}\left[\mathbf{E}\left[(\tau_t - \tau^*)\frac{1}{b_{t+1}}\tilde{h}(\tau_t) \middle| \mathcal{F}_t\right]\right] + \mathbf{E}\left[\mathbf{E}\left[\left(\frac{\tilde{h}(\tau_t)}{b_{t+1}}\right)^2 \middle| \mathcal{F}_t\right]\right] \\
&= \beta_t + 2\mathbf{E}\left[\mathbf{E}\left[\frac{\tau_t - \tau^*}{b_{t+1}}\widehat{h}(\tau_t) \middle| \mathcal{F}_t\right]\right] \\
&\qquad + 2\mathbf{E}\left[\mathbf{E}\left[\frac{\tau_t - \tau^*}{b_{t+1}}(\tilde{h}(\tau_t) - \widehat{h}(\tau_t)) \middle| \mathcal{F}_t\right]\right] + \mathbf{E}\left[\mathbf{E}\left[\left(\frac{\tilde{h}(\tau_t)}{b_{t+1}}\right)^2 \middle| \mathcal{F}_t\right]\right] \\
&= \beta_t + 2\mathbf{E}\left[\frac{\tau_t - \tau^*}{b_t + 2}h(\tau_t)\right] + 2\mathbf{E}\left[\mathbf{E}\left[\frac{\tau_t - \tau^*}{b_{t+1}}(\tilde{h}(\tau_t) - \widehat{h}(\tau_t)) \middle| \mathcal{F}_t\right]\right] + \mathcal{O}\left(\mathbf{E}\left[\frac{1}{b_t^2}\right]\right) \quad (27) \\
&= \beta_t - 2\mathbf{E}\left[\frac{\tau^* - \tau_t}{b_t + 2}h(\tau_t)\right] + \mathbf{E}\left[\frac{\tau_t - \tau^*}{b_t + 2}\|\eta - g_t\|_1 + \mathcal{O}\left(\frac{1}{b_t^2}\right)\right] + \mathcal{O}\left(\mathbf{E}\left[\frac{1}{b_t^2}\right]\right) \quad (28) \\
&= \beta_t - 2\mathbf{E}\left[\frac{\tau^* - \tau_t}{b_t + 2}h(\tau_t)\right] + \mathbf{E}\left[\frac{\tau_t - \tau^*}{b_t + 2}\|\eta - g_t\|_1\right] + \mathcal{O}\left(\mathbf{E}\left[\frac{1}{b_t^2}\right]\right)
\end{aligned}
$$

where (26) follows from the update rule defined in (23), and (27) follows from the fact that $\tilde{h}(\tau_t)$ is bounded and $1/(b_t + 2) \leq 1/b_{t+1} \leq 1/b_t$. Finally, Lemma 11 is applied in (28).

As a next step, we show that

$$
\begin{aligned}
\sum_{t=1}^{\infty} \mathbf{E}\left[\frac{\tau_t - \tau^*}{b_t + 2}\|\eta - g_t\|_1\right] &\leq \sum_{t=1}^{\infty} \mathbf{E}\left[\frac{\|\eta - g_t\|_1}{b_t + 2}\right] \\
&\leq \sum_{t=1}^{\infty} \sqrt{\mathbf{E}\left[\frac{1}{(b_t + 2)^2}\right]} \sqrt{\mathbf{E}\left[\|\eta - g_t\|_1^2\right]} \qquad (29) \\
&\leq \sum_{t=1}^{\infty} \mathcal{O}\left(\frac{1}{t}\right) \sqrt{\mathbf{E}\left[\|\eta - g_t\|_1^2\right]} \qquad (30) \\
&\leq \sum_{t=1}^{\infty} \mathcal{O}\left(\frac{1}{t}\right) \sqrt{\mathbf{E}\left[2\|\eta - g_t\|_1\right]} \qquad (31) \\
&= \sum_{t=1}^{\infty} \mathcal{O}\left(\frac{1}{t^{1 + \lambda/2}}\right) < \infty \ . \qquad (32)
\end{aligned}
$$

where the Cauchy-Schwarz inequality is applied in (29), and (30) follows from Lemma 8 and, (31) follows from the fact that $\|\eta - g_t\|_1 \leq 2$. Finally, (32) follows from the assumption regarding the online learner, namely, there exists an $\lambda > 0$, such that

$$
\mathbf{E}\left[\int_{\boldsymbol{x} \in \mathcal{X}} |\eta(\boldsymbol{x}) - g_t(\boldsymbol{x})| \, \mathrm{d}\mu(\boldsymbol{x})\right] = \mathcal{O}\left(t^{-\lambda}\right)
$$

Therefore, in this case, $\beta_{t+1}$ can be decomposed in a similar way like in (19). Plugging this result into (20), the rest of the proof is analogous to the proof of Theorem 3. $\qquad \square$

## D  Hyperparameters of the online learners

The regularized version of the Logistic Regression was trained as it is introduced in [15]. This model has three hyperparameters the initial learning rate $\eta$, and the regularization coefficients for L1 and L2 penalties that are denoted by $\lambda_1$ and $\lambda_2$ respectively. The PEGASOS learning method has a single hyper-parameter $\lambda$, which controls the step size of the parameter update. Finally, the only hyperparameter of PERCEPTRON algorithm is the learning rate which we denote by $\gamma$. The hyperparameters and their values on which they were optimized are summarized in Table 2.

Table 2: The hyperparameters of the learning methods and their ranges used in hyperparmater optimization.

| Learner | hyperparameter | Validation range |
|---|---|---|
| LR | $\eta$ | $\{0.1, 0.01, 0.001, 0.0001\}$ |
|  | $\lambda_1$ | $\{0.1, 0.01, 0.001, 0.0001\}$ |
|  | $\lambda_2$ | $\{0.1, 0.01, 0.001, 0.0001\}$ |
| PEGASOS | $\lambda$ | $\{0.5, 0.4, 0.3, 0.2, 0.1, , 0.05, 0.03, 0.01, 0.005, 0.001, 0.0001\}$ |
| PERCEPTRON | $\gamma$ | $\{10.0, 5.0, 3.0, 1.0, 0.5, 0.3, 0.1, 0.05, 0.01, 0.005, 0.001, 0.0001\}$ |

## E  Optimal threshold $\tau^*$ with Beta and uniform distributions as $\nu(\cdot)$

**Example 12.** *Assume that $\nu(\cdot)$ is uniform on $[0, 1]$. Then, according to (8), $h(\tau)$ is given as follows:*

$$h(\tau) = 1/2 - \tau^2/2 - \tau(1 - \tau) - \tau/2$$
$$= \tau^2 - 3\tau + 1$$

*Setting $h(\tau) = 0$ and noting that the solution has to be in $[0, 1]$, the optimal threshold is $\tau^* = (3 - \sqrt{5})/2$, whence $F(\tau^*) = 3 - \sqrt{5} \approx 0.76$.*

**Example 13.** *Assume that $\nu(\cdot)$ is the Beta density function with parameters $\alpha$ and $\beta$: $\nu(p) = p^{\alpha-1}(1 - p)^{\beta-1}/B(\alpha, \beta)$, where $B(\alpha, \beta) = \Gamma(\alpha + \beta)/(\Gamma(\alpha) + \Gamma(\beta))$ and $\Gamma(\cdot)$ is the Gamma function. The cumulative distribution function of the Beta distribution is denoted by $F(x\,;\alpha, \beta)$ and defined as $F(x\,;\alpha, \beta) = B(x\,;\alpha, \beta)/B(\alpha, \beta)$, where $B(x\,;\alpha, \beta) = \int_0^x t^{\alpha-1}(1 - t)^{\beta-1}\,\mathrm{d}t$. Thus, $h(\tau)$ can be written as follows:*

$$h(\tau) = \frac{1}{B(\alpha, \beta)} \int_\tau^1 p^\alpha (1 - p)^{\beta-1}\,\mathrm{d}p - \tau(1 - F(\tau\,;\alpha, \beta)) - \tau \frac{\alpha}{\alpha + \beta}$$
$$= \frac{B(\alpha + 1, \beta)}{B(\alpha, \beta)}(1 - F(\tau\,;\alpha + 1, \beta)) - \tau(1 - F(\tau\,;\alpha, \beta)) - \tau \frac{\alpha}{\alpha + \beta}$$
$$= \frac{\tau(1 - \tau)B(\alpha + 1, \beta)}{B(\alpha, \beta)} - \tau \frac{\alpha}{\alpha + \beta}$$

*Noting that $\tau^* = 0$ can be excluded, a simple calculation yields $\tau^* = 1 - \frac{\alpha}{\alpha+\beta}\frac{B(\alpha,\beta)}{B(\alpha+1,\beta)}$.*

## F  Synthetic Data

We sampled instance vectors from a two-dimensional Gaussian with mean $(1, 1)$ and the identity as covariance matrix. The class conditional probability for an instance $\boldsymbol{x} = (x_1, x_2)$ is computed by a logistic function as follows: $\eta(\boldsymbol{x}) = \mathbf{P}(Y = 1|X = \boldsymbol{x}) = \frac{1}{1+\exp(-x_1+x_2+3))}$. Figure 2 (a) and (b) show a sample dataset consisting of 30000 instances and the contour plot of their posteriors probabilities, respectively.

We run the OFO algorithm with logistic regression without regularization as online learner on 30000 random examples $\mathcal{D}_N$ in order to satisfy the required consistency condition. The initial learning rate was set to 0.01 and no regularization was applied here. Since the precise posteriors are known, the

Figure 2: Experiments with synthetic data. (a) The data points sampled from a two-dimensional normal distribution. The blue line indicates the Bayes optimal decision surface for which $\mathbf{P}(Y = 1 \mid X = \boldsymbol{x}) = 1/2$. The red points represent the positive instances. (b) The contour plot of the posterior probabilities $\mathbf{P}(Y = 1 \mid X = \boldsymbol{x})$. (c) The solid and dashed black lines show the F-measure and the validated threshold, respectively, in batch mode (note that $F(\tau^*) = 2\tau^*$). The red lines show the online F-score and the threshold found by OFO with logistic regression. Similarly, the solid lines represent the F-score and the dashed one the threshold.

validated threshold and F-measure were computed as a baseline. These are obtained by optimizing $F(\tau; \eta(\cdot), \mathcal{D}_N)$ as described in Section 2. The dashed black lines in Figure 2 (c) show the validated threshold (below) and the F-measure (above) found (note that $F(\tau^*) = 2\tau^*$). The online F-measure and thresholds found by OFO are plotted with solid and dashed red lines, respectively. As the results show, the online algorithm converges to the validation-based approach in terms of both F-measure and threshold, which in accordance with our theoretical results. All results are obtained by averaging over 100 repetitions.

# G   Online F-scores obtained by OFO algorithm on benchmark datasets

Figure 3: Online F-scores obtained by OFO algorithm with various online learners on benchmark datasets