[Reviews · NeurIPS 2015]

Submitted by Assigned_Reviewer_1

This paper proposes an algorithm that can optimize the F-Measure in an online fashion. To do that, it relies on an online learner to estimate the posterior which is utilized at each updating. The algorithm has been proved theoretically to be able to converge to the optimal threshold computed based on batch learning. Empirical evaluation shows consistent results. The experimental results on sensitivity towards three different online learners are shown. The paper is clearly presented and organized. Also, the proposed algorithm can be of practical use.
Summary: This paper proposes an algorithm that can optimize the F-Measure in an online fashion. The effectiveness of the proposed algorithm has been evaluated both theoretically and empirically with promising results.

Submitted by Assigned_Reviewer_2

The problem of learning binary classifiers optimizing non-decomposable performance metrics has recently gained a lot of traction; this paper considers the popular F-measure specifically, and proposes an algorithm for optimizing F-measure in the online learning setting. The authors give a formal consistency analysis for the online algorithm. Experimental results on fairly large real-world datasets are presented.

The main idea in the paper -- and the online algorithm itself -- is neat and simple. It is easy to understand and more or less easy to implement (except for the caveat of estimating a model for the class-conditional probabilities in each round, which can be quite tricky). The consistency analysis is also fairly novel. Though, I'm not quite sure if the way the guarantee is presented in Theorem 3 (and 5) is precise. Perhaps it would be clearer to make the regret, which in this case is F((y_1, ... , y_t), (\hat{y}_1,...,\hat{y}_t)) - F(\tau^*)), explicit in the statement (which the authors do mention later in Section 6).

Recently, [9] introduced a framework for optimizing & analyzing non-decomposable measures in the online setting. It is not quite convincing from the discussion in Section 6, how exactly the analysis of OFO is stronger than that in [9]. In fact, the framework in [9] based on a certain per-round non-decomposable loss is quite intuitive. In particular, [9] also includes convergence rates for regret. It is also surprising that the method in [9] is not compared to in experiments (at least a justification is needed otherwise).

The other concern is with respect to the organization of the paper itself. The main idea/contribution which is Algorithm 1 and its analysis appear much later in the paper. Sections 2 and 3 are redundant for the most part. In particular, I fail to see the point of Section 3 in the context of this paper and its key contribution --- at best, it can be summarized in a Background section that combines the key ideas in Sections 2 and 3. This way, the readers can get to Section 4 (& 5), which is the focus of this paper, without distractions (and possibly confusions). The authors even had to move some useful results on Synthetic data to the Appendix -- I suggest including this in the main paper.

Responses are satisfactory. Raising my score to Accept.

Summary: The paper considers the problem of optimizing F-measure in the online setting. An online algorithm, that is easy to understand and implement, is proposed and its consistency is analyzed. Experimental results showing conformance to theory is presented. The paper does make sufficient contributions, but I've some concerns with evaluation and presentation/development of content.

Submitted by Assigned_Reviewer_3

Quality: - This paper is technically sound and the supplemental file contains the detailed proofs and extensive experiments.

The discussion in Session 6 is interesting

and insightful, especially the argument that F-measure itself is an aggregate measure which thus makes no sense to analyze the regret bound in general online learning.

- In line 196-197, although $\hat{h}(\tau)$ is an unbiased estimate of $h(\tau)$, what is the variance of $\hat{h}(\tau)$ w.r.t. the data distribution? Alternatively, it is suggested to add some discussions on other estimated functions of $h(\tau)$. - Moreover, in line 268-269, it is stated that the noisy estimation of $h(.)$ depends on the labels. How can the decomposition step help to handle this undesired effect?

- In line 286, it is better to clarify the assumptions that are required on the online learner.

Clarity: - This paper is well written and clearly organized, although there are still some minor issues.

The overall idea of this paper is easy to follow. - For example, the second "-" sign in Eq.(5) should be a "+" sign.

- There seems to be an error in the most right hand side of Eq.(1).

- In line 155, what does "P" above the arrow denote? It should be clarified first before its usage. - In line 65-66, "positive a negative" should be " positive and negative". - In line 7 of the pseudo-code, it is better to add some notes that $a_t$ and $b_t$ can be calculated by Eq.(9). - In line 251-252, what do $F_t$-measurable and filtration mean? It is better to add more explanation on them. - In line 296, " conference rate" may be " convergence rate".

Originality: In this paper, the authors study the online optimization problem of a non-decomposable metric, the F-score, which is less explored in the community. The reviewer thinks the idea in this paper is novel.

Significance: Since there are still several nice properties of the four fundamental quantities (true/false positive/negative rate), more exploration based on this paper is expected. Moreover, since F-score function is neither convex nor concave and has the decomposition nature, there are still more challenging open problems on F-measure optimization. The reviewer thinks this paper can raise further research interest in the online learning community.
Summary: In this paper, the authors propose an online algorithm to threshold a given classifier for the purpose of F-measure maximization and prove the convergence of the online F-scores obtained by the proposed algorithm to the optimal F-score.

Extensive experiments are conducted to verify its convergence properties in the theoretical analysis.

Submitted by Assigned_Reviewer_4

The paper studies the problem of training classifiers that maximize the F-measure in the online setting, which is defined as the ratio of two linear functions and thus, unlike the traditional loss functions, is non-decomposable. Most existing algorithmic approaches of optimizing F-measure are in the batch setting and solve the problem by finding the optimal threshold of probabilistic classifiers on a validation set. This paper proposes an online learning algorithm to find the optimal threshold and proof consistency results under various assumptions. This authors also include comprehensive experiments (including the appendix) to demonstrate the performance of the proposed method.

Generally, the paper is well written and I like the presentation of the proof the Theorem 3. The theoretical results seem to be solid and experimental section is fairly convincing.

This is not my area of expertise, so I'm less certain about the novelty of the technical contribution comparing to the existing literature on F-measure maximization. That been said, I like that this paper has a balanced combination of theory/algorithm/experiments.

Minor comments:

In the Discussion section (Section 6), it would be nice to expand the comparisons with [9]. In terms of AUC maximization in the online setting, there are two relevant papers on the analysis of the generalization performance.

1. Wang et al., Generalization bounds for online learning algorithms with pairwise loss functions, COLT 2012

2. Kar et al., On the generalization ability of online learning algorithms for pairwise loss functions, ICML 2013

Line 454: extra space in "consistency"

Summary: This paper proposes an online learning algorithm to optimize the F-measure with a nice combination of theory, algorithms and experiments.

Author Feedback
Author rebuttal: We would like to thank the reviewers for the positive feedback!

Rev 2:

---> ... variance of $\hat{h}(\tau)$ w.r.t. the data distribution?

Computing variance would be very helpful -- but also quite challenging -- as it might be a key point for computing convergence rates (and not only consistency). In particular, the variance can be used to compute high probability confidence intervals for h(\tau), which is usually a basic ingredient in convergence analysis. Finally, based on Prop 9, we know that h(\tau) is not "too flat" anywhere. However, coming up with a reasonable variance estimate with respect to the data is not easy. We'll make this point explicit in the final version.

---> ... noisy estimation of $h(.)$ depends on the labels ... decomposition step ...

According to Prop 2, our update rule is very similar to the original Robbins-Monro (RM) algorithm. But the coefficient and the noisy estimation of h(\tau) both depend on the currently observed instance (x_{t+1},y_{t+1}) which is not known at time t. Therefore, they are not independent and the original analysis of RM cannot be applied directly. Yet, the decomposition allows us to rewrite the update rule (in expectation) such that all terms only depend on the observations up to time t (see Lemma 7), whence they are fixed values with respect to the filtration \cal{F}_{t}. Thus, the analysis of RM could be adapted to our setup.

---> ... clarify the assumptions that are required on the online learner.

Thank you for pointing this out. We shall add a short discussion about the assumption in Theorem 5 and, in particular, about how it relates to the assumptions underlying related results in the literature (e.g. [9]).

Rev 3:

Elaborating on the relation between online F-measure and AUC optimization is a good idea. We are going to make this point explicit in the paper with the proposed citations that seem to be relevant. Yet, we like to point out that, from an algorithmic point of view, AUC optimization is quite different from (online) F-measure optimization.

Rev 4:

---> Perhaps it would be clearer to make the regret ... explicit in the statement ...

We like to emphasise that we do not investigate any regret. Instead, we showed that F((y_1, ... , y_t), (\hat{y}_1,...,\hat{y}_t)) - F(\tau^*)) will approach zero with probability one as t goes to infinity. Due to its non-decomposable nature, it is even unclear how a reasonable regret could be defined for F-measure. This point is discussed in Sec 6.

---> [9] introduced a framework for optimizing & analyzing non-decomposable measures ...

Elaborating on the relationship between [9] and our approach, both on a formal and empirical level, is indeed an interesting issue, and we definitely plan to do so in an extended version of this paper. Having said that, such a comparison (like the unification of the adversarial and stochastic multi-armed bandits in general) is not at all straightforward.

In fact, [9] shares commonalities with our setting but also exhibits important differences. For example, thanks to our assumption of an underlying distribution on instance/label pairs, our model ensures that the F-measure (7) is maximized by thresholding the optimal classifier at some \tau^*. Moreover, our analysis shows that the performance of OFO converges to this maximum, F(\tau^*). These properties do not seem to be shared by the algorithm in [9].

Indeed, it is not clear whether the setting in [9] guarantees the existence of an optimal threshold classifier, and in any case, it pursues a different goal. Introducing a surrogate model, it aims at converging to the best linear estimator of the F-measure in the structural SVM sense: given x_1, ..., x_n and y_1, ..., y_n, find the w that gives the tightest estimate of F_1(y_1, ..., y_n; y'_1, ..., y'_n) in the form of \sum_i y_i w^T x_i - \sum_i y'_i w^T x_i in the worst case sense (i.e., for the "worst" y'_1, ..., y'_n). Therefore, as the F-measure is not flat, this goal can be very different from ours. This also explains why the notion of regret makes perfect sense in their work but not in ours (see the first paragraph of Sec 6).

As for the experiments, comparing to the approach based on threshold-validation as a baseline is most natural for us, because this (batch) version yields the optimal (offline) solution to the problem that we try to solve in an online manner (and has similar consistency properties). Compared to this, the approach of [9] optimizes for a task that seems to be much more different from ours.

We'll try to make the relationship to [9] clearer and put parts of this discussion in the final version of the paper.

Organization: We shall merge Sec 2 and 3 and shorten them. We believe the technical content of Sec 3 helps to understand our consistency result in Sec 5. The synthetic result was deferred to the supplementary due to lack of space, but we shall put it back in the final version.